# Changes in Faecal Short-Chain Fatty Acids after Weight-Loss Interventions in Subjects with Morbid Obesity

**DOI:** 10.3390/nu12030802

**Published:** 2020-03-18

**Authors:** Per G Farup, Jørgen Valeur

**Affiliations:** 1Department of Research, Innlandet Hospital Trust, N-2381 Brumunddal, Norway; 2Department of Clinical and Molecular Medicine, Faculty of Medicine and Health Sciences, Norwegian University of Science and Technology, N-7491 Trondheim, Norway; 3Unger-Vetlesen Institute, Lovisenberg Diaconal Hospital, N-0440 Oslo, Norway; jorgen.valeur@lds.no

**Keywords:** obesity, short-chain fatty acids, bariatric surgery, weight-loss, faecal microbiota

## Abstract

The gut microbiota and their metabolites, e.g., short-chain fatty acids (SCFA), are associated with obesity. The primary aims were to study faecal SCFA levels and the changes in SCFA levels after weight-loss interventions in subjects with obesity, and secondarily, to study factors associated with the faecal SCFA levels. In total, 90 subjects (men / women: 15/75) with a mean age of 44.4 (SD 8.4) years, BMI 41.7 (SD 3.7) kg/m^2^ and morbid obesity (BMI > 40 or > 35 kg/m^2^ with obesity-related complications) were included. Faecal SCFA and other variables were measured at inclusion and after a six-month conservative weight-loss intervention followed by bariatric surgery (Roux-en-Y gastric bypass or gastric sleeve). Six months after surgery, the total amount of SCFA was reduced, the total and relative amounts of the main straight SCFA (acetic-, propionic-, and butyric- acids) were reduced, and the total and relative amounts of branched SCFA (isobutyric-, isovaleric-, and isocaproic- acids) were increased. The changes indicate a shift toward a proteolytic fermentation pattern with unfavorable health effects. The amount of SCFA was associated with the diet but not with metabolic markers or makers of the faecal microbiota composition. Dietary interventions could counteract the unfavorable effects.

## 1. Introduction

The gut microbiota and their metabolites, e.g., short-chain fatty acids (SCFA), have health-related effects and have been associated with a wide range of disorders [1,2]. Obesity with comorbidities is one of these microbiota-associated disorders, although a causal relationship has not been documented in humans [3,4,5]. The microbiota and the metabolites might be both health-promoting and health-damaging. All the individual SCFA are present under physiological conditions and play different roles. An imbalance in the pattern, e.g., in the saccharolytic fermentation characterized by an increase in the main straight SCFA (acetic-, propionic, and butyric- acids) versus the proteolytic fermentation, characterized by an increase in the branched SCFA (isobutyric- isovaleric-, and isocaproic- acids), may signify alterations in the microbial functions that may be associated with either gut health or disease [1,2,6,7,8,9]. Knowledge of faecal SCFA in subjects with morbid obesity and the changes after a combined conservative and surgical intervention is limited [4,10,11,12,13,14]. An unbalance in the SCFA pattern before or after weight-reducing treatment might have unfavorable health effects that necessitate interventions.

The primary aims were to study faecal SCFA in subjects with morbid obesity and the changes in SCFA after a combined conservative and surgical treatment, and secondarily, to study associations between SCFA and the diet, the faecal microbiome composition and some metabolic and inflammatory biomarkers (HbA1c, CRP, and s-zonulin).

## 2. Materials and Methods

### 2.1. Study Design

Consecutive subjects with morbid obesity referred to Innlandet Hospital Trust, Gjøvik, Norway for evaluation of bariatric surgery were evaluated for inclusion in this prospective cohort study. After inclusion (T 1) and before bariatric surgery, the subjects completed a six-month conservative treatment period. This is standard procedure, and the conservative weight loss intervention helps the subjects to adapt to lifestyle changes. There was a follow-up visit six months after surgery (T 2).

### 2.2. Inclusion Criteria

Subjects 18–65 years of age with morbid obesity (defined as BMI > 40 kg/m^2^ or > 35 kg/m^2^ with obesity-related complications) were available for inclusion. Subjects with previous major gastrointestinal surgery, organic gastrointestinal disorders, alcohol and drug abuse, major psychiatric disorders, and serious somatic disorders not related to obesity were excluded.

### 2.3. Interventions

The conservative weight-loss intervention period started with three one-hour-long visits separated by one week; consulting a nurse, a nutritionist and a physician. The participants were given individualized dietary advice, physical activity programs and information about the operation and consequences of the operation. Some weeks later, they participated in weekly group meetings for seven weeks chaired by nurses, nutritionists, surgeons and a psychologist. The last three weeks before surgery, they followed a strict “crispbread diet” containing 4200 kJ of energy [15].

Three experienced surgeons performed bariatric surgery with one of two standard methods, either Roux-en-Y gastric bypass or gastric sleeve, chosen at the surgeons’ discretion [16,17].

### 2.4. Variables

The following variables were collected at inclusion (T 1) and six months after bariatric surgery (T 2):

Demographic and anthropometric data including age (years), gender (male/female) smoking habits (daily smoking/ not daily smoking), height (m), body weight (kg) and body mass index (BMI; kg/m^2^), and present and previous diseases.

A blood sample was analyzed for a range of haematological and biochemical variables including C-reactive protein (CRP, normal range < 3.0 mg/L; a marker of inflammation), HbA1C (normal range < 5.6%; a marker of metabolic health) and serum zonulin (normal range < 38 ng/mL; a marker of gastrointestinal permeability). CRP and HbA1C were analyzed with a Cobas c501 instrument with the reagents CRPL3 and Tina-quant HbA1C (Roche Diagnostics GmbH, 68305 Mannheim, Germany), and s-zonulin was measured with an ELISA kit (Immundiagnostik, 64625 Bensheim, Germany).

Dietary habits were assessed with a self-reported food frequency questionnaire (FFQ) constructed and validated by the University of Oslo [18]. The University of Oslo calculated the daily intake of nutrients and supplements including non-nutritive sweeteners (NNS) based on the Norwegian food composition table [19]. One unit of NNS was defined as 100 mL of beverages sweetened with NNS, or two tablets/teaspoons of NNS.

The faecal material for the analyses of the microbiota and SCFA was collected by the subjects at home in a “Sample Collection Kit” provided by Genetic Analysis AS, Oslo, Norway, the company that analyzed the microbiota composition, and handled according to their recommendations: *“The kit is designed to ensure hygienic and easy sampling of the faecal material and can be performed at home. No additives are required. The sample should be stored in room temperature and reach the laboratory within 5 days”* [20]. Upon arrival to the hospital, the samples were immediately stored at −80 °C and later transported in batches for the analyses of the microbiota. Afterwards, the samples were transferred to Unger-Vetlesen Institute, Oslo, Norway, for the analyses of SCFA. All the time from arrival to the hospital to the last analyses had been performed, the samples were stored at −80 °C.

The faecal microbiota composition was analyzed with the commercially available, CE marked, US and European patented, GA-map™ dysbiosis test (Genetic Analysis AS, Oslo, Norway) [21,22]. The test reports the degree of dysbiosis as Dysbiosis Index (DI; range 1–5). Values above 2 indicate a microbiota composition that differs from a reference population. Also, the relative abundance of 39 bacteria at different taxonomic levels are reported as score -3 to 3 relative to the reference population. Twenty-four of the bacteria were from the phylum Firmicutes and eight from Bacteroidetes. The relative abundance of bacteria from the phyla Firmicutes and Bacteroidetes were calculated as the mean of the relative scores from the bacteria in these phyla. Note that the bacteria measured with the actual method do not represent the entire phyla but only parts of the phyla.

Faecal short-chain fatty acids (SCFA) were analyzed as described by Zijlstra et al. and modified by Høverstad et al. [23,24]. The distillate was analyzed with gas chromatography and quantified by using internal standardization. Flame ionization detection was employed. The total amount of all SCFA and the amount of acetic-, propionic-, butyric-, isobutyric-, valeric-, isovaleric-, caproic-, and isocaproic- acids was measured and expressed in mmol/kg wet weight. Some subjects had two analyses of faecal SCFA, the one that was planned six months after surgery and an extra one 12 months after surgery. Some had a test only after 12 months. In subjects with two analyses, there were no significant differences between the results. Therefore, in subjects with only one measurement, the results of the available test 6 or 12 months after surgery were used. In subjects with two analyses, the mean values of the two tests were used.

### 2.5. Statistics

A linear mixed model was used for the majority of the analyses. The dependent variables appear in the result section. Subject was the random effect. Explanatory variables were the point of time (a two-level categorical covariate), the mean of age and gender, type of operation, and various variables presented in the result section. When appropriate, interaction analyses were performed. Associations between the changes in SCFA and changes in nutrients, biological markers and the microbiota composition were analyzed with linear regression adjusted for age and gender. The analyses were performed with IBM SPSS Statistics for Windows, version 25.0 (IBM Corp., Armonk, NY, USA). P-values < 0.05 were judged as statistically significant. The sample size was fixed by the available study population and no power calculation was performed during the planning of the study.

### 2.6. Ethics

The study was approved by the Regional Committee for Medical and Health Research Ethics South-East Norway (reference 2012/966) and conducted in accordance with the Declaration of Helsinki. All participants gave written informed consent before inclusion in the study.

## 3. Results

### 3.1. Subjects

Out of 239 subjects available for inclusion, 80 refused to participate, 7 with previous or present somatic disorders were erroneously included and later excluded, 21 had no operation, and 41 did not provide faecal samples. In total, 15 (17%) men and 75 (83%) women with a mean age of 44.4 (SD 8.4) years and BMI 41.7 (SD 3.7) kg/m^2^ were included in the analyses and 80 had a follow-up visit six months after surgery. At inclusion, BMI was higher in men than in women (difference of 2.99 kg/ m^2^ (CI: 0.23 to 3.76; *p* = 0.027)) and decreased 0.15 kg/m^2^ per year as age increased (CI: 0.07 to 0.23; *p* < 0.001). The mean reduction in BMI after the interventions was 12.70 kg/m^2^. (CI: 12.03 to 13.38; *p* < 0.001). Roux-en-Y gastric bypass was performed in 73 (81%) and gastric sleeve in 17 (19%). The weight-loss was significantly higher in subjects operated with Roux-en-Y gastric bypass than in those operated with gastric sleeve, difference 1.73 kg/m^2^ (CI: 0.03 to 3.42; *p* = 0.046).

### 3.2. Short-Chain Fatty Acids

Total SCFA levels were reduced after treatment. The absolute and relative amounts of all the SCFA at inclusion and the changes after treatment are given in Table 1. The dependent variables were not associated with the type of operation. The major straight SCFA (acetic-, propionic-, and butyric- acids) changed principally in the same way, as did the branched SCFA (isobutyric-, isovaleric-, and isocaproic- acids). Therefore, in the further analyses they were considered as two groups. The absolute and relative amounts of the straight SCFA were reduced and the absolute and relative amounts of the branched SCFA were increased.

### 3.3. Nutrients, Blood Tests, Type of Surgery and Faecal Microbiota Composition

The energy intake was significantly reduced after the weight-loss interventions. Except for an increase in the relative energy amount of protein and fiber, there was a reduction in all absolute and relative amounts of the nutrients after treatment. The markers of inflammation (CRP), metabolic syndrome (HbA1C) and gut permeability (zonulin) normalized. There was a change in the faecal microbiota composition towards dysbiosis and an increase in the relative amount of Firmicutes. Table 2 gives the details. The type of bariatric surgery was not significantly associated with the changes.

### 3.4. Associations between SCFA Levels and Other Variables

There were significant positive associations between total SCFA and the sum of the straight SCFA and the intake of energy, protein, fat, and starch, but no significant associations with the blood biomarkers and the faecal microbiota composition markers. Table 3 gives the details. Type of surgery was not significantly associated with the SCFA levels, and there were no significant interactions with the point of time.

Out of the associations between changes in total, straight, and branched SCFA on one side (dependent variables) and changes in the nutrients, biological markers and the microbiota composition on the other side, the only significant association was between the change in branched SCFA and the change in the intake of starch. (B: −0.12 (CI: −0.022 to −0.002); partial correlation: −0.344; *p* = 0.019. All results are given in Table 4.

## 4. Discussion

The main findings were the significant changes in faecal SCFA levels after a conservative weight-loss intervention followed by bariatric surgery. Six months after surgery, the total amount of SCFA was reduced, the absolute and relative sum of the main straight SCFA (acetic-, propionic-, and butyric- acids) was reduced, and the absolute and relative sum of the branched SCFA (isobutyric-, isovaleric-, and isocaproic- acids) was increased. The results are in accordance with reports of other conservative and surgical treatment alternatives [11,12,25]. The design renders the separation of the effects of the two interventions impossible. Other studies have shown that the effect of weight loss on inflammatory biomarkers (e.g., neopterin [26]) and gastrointestinal permeability (submitted by one of the authors) is unrelated to the changes in BMI and could be even higher after conservative rather than surgical treatment of obesity. The separation of the effects of the two interventions is, however, less important than the overall effect since the procedure is a standard treatment combination for subjects with morbid obesity. The reduction of straight SCFA and increase in branched SCFA indicates reduced saccharolytic and increased proteolytic fermentation, respectively [2,9]. A review concludes that the faecal concentrations of the major straight SCFA are elevated in subjects with obesity [4]. Since valid reference values were unavailable for the method used for the analyses, it is unknown if the major straight SCFA were elevated before treatment and then normalized, or were normal and reduced to subnormal amounts after treatment. Possible causes of the changes in SCFA are changes in the diet, the gut microbiota and their host.

There was a significant reduction in food intake. The absolute and relative amounts of all parts of the diet were reduced, except for a significant increase in the relative amount of protein and a minor increase in fiber. Changes in the diet, in particular polysaccharides and proteins, alter the microbiota and their fermentation products such as SCFA [2,8,9,27,28,29,30]. There were significant associations between the dietary intake of energy, protein, fat, and starch on one side and the amount of total and straight SCFA, and a significant negative association between the changes in the intake of starch and branched SCFA. The study confirms the associations between the diet and SCFA. The negative association between the changes in the intake of starch and branched SCFA shows the importance of a carbohydrate-rich diet for the reduction of the proteolytic fermentation. The methods measuring the dietary intake and SCFA were judged as valid and reliable. The paper by Tremaroli et al. reports similar changes in SCFA after bariatric surgery and concludes that the changes were not a consequence of the dietary consumption [12]. NNS, which was used in high amounts by a substantial proportion of the participants, were not associated with changes in SCFA. An association was anticipated since NNS induce marked changes in the gut microbiome [31,32,33,34]. Separate analyses during the conservative weight-loss period with primarily dietary restrictions could perhaps have shown more explicit associations between changes in the diet and changes in the SCFA. The surgical procedures probably have different and more substantial impacts on the SCFA than the diet. There were, however no associations between the surgical methods and SCFA levels.

At inclusion, the faecal microbiota composition showed a minor deviation from a reference population (a slight degree of dysbiosis) and a further deviation after surgery. Changes in the gut microbiome composition have been reported in several studies in obese subjects, but there is no agreement concerning the type of deviation and causal relations [3,5]. Changes of the microbiota composition after conservative and surgical weight-loss have also been reported [10,13,14]. The increasing degree of dysbiosis after treatment indicates that the treatment does not reset the microbiota, in fact it is the opposite [35]. The method used for the analyses of the microbiota composition did not allow precise characterization of the microbiota and the changes of the microbiota since the method measured only 39 bacteria at different taxonomic levels. The same changes were in a previous study based on the same material judged as unfavorable (“bad” dysbiosis) [34]. In contrast to the review by Wagner el al., this study showed a significant increase in the relative amount of the phylum Firmicutes and a non-significant reduction in Bacteroidetes [5]. An abundance of Firmicutes and a high Firmicutes/Bacteroidetes ratio have been associated with obesity and judged as unfavorable [36,37,38]. If correct, the changes observed in this study are thus unfavorable. Note that the phyla do not include the complete phyla but only a selection of the microbes present in the phyla. A better characterization of the microbiome composition seems necessary to show associations between the microbiome composition and faecal SCFA levels.

The treatment has several other important health-related impacts on the subjects, such as metabolic and inflammatory changes, changes in physical activity, use of drugs, and gastrointestinal malabsorption and permeability (factors that have an impact on the gut microbiota and their function). In this study, these factors were limited to the study of CRP (a marker of inflammation), HbA1c (a marker of metabolic syndrome), and zonulin (a marker of intestinal permeability) which showed significant normalization after treatment but were not associated with the amount of, or changes in, SCFA levels. In all, the study gives no clear causative explanation of the changes in SCFA levels. The reduction of the total SCFA levels and the major straight SCFA (indicating reduced saccharolytic fermentation) could be due to a reduced intake of nutrients and carbohydrates [29]. The increase in branched SCFA levels (indicating increased proteolytic fermentation) indicates an increase in proteins in the colon that could be due to the increase in the relative amount of protein in the diet or minor protein malabsorption [39].

The microbial fermentation metabolites are markers for health, but the impact of these products on human health is complex, and the clinical consequences of the changes in faecal SCFA levels are not fully understood [1,9]. Low SCFA levels increase energy intake and reduce energy expenditure [40]. The saccharolytic fermentation with the production of acetic-, propionic-, and butyric acids has some health-promoting effects relating to fatty acids, glucose, and cholesterol metabolism, mineral absorption, the regulation of immune and inflammatory responses, as well as being a source for colonocyte energy and tissue repair including the gut barrier function, and having anti-obesogenic, antioxidant and anticancer effects [7,8,28,41,42]. The proteolytic fermentation with an increase in branched SCFA levels is associated with the production of harmful metabolites such as ammonia, phenols and hydrogen sulphides that have clinical relevance for disorders like irritable bowel syndrome, inflammatory bowel diseases and cancer [40,43,44]. In all, the observed alteration from a saccharolytic to proteolytic fermentation after treatment for morbid obesity seems detrimental. Although the clinical relevance is uncertain, a recommendation of a carbohydrate-, fiber-, and polysaccharide-rich diet aiming at a shift toward a saccharolytic fermentation seems reasonable.

The study included consecutive and unselected subjects with morbid obesity referred to the public obesity unit in the region and was performed as part of the daily routine. Data on comorbidity, complications and pharmacotherapy were incompletely registered. The subjects performed a standard combined conservative and surgical intervention. The results were limited to this group in which the majority were females. The validity of the results for men might be reduced, the validity for subjects with less severe obesity is unknown, and the changes after only conservative or surgical treatment might differ [10]. SCFA measured in faeces do not reflect the colonic SCFA production since the majority of SCFA are absorbed within the colon and only a minor proportion (5%–10%) are excreted in faeces. Faecal SCFA are nevertheless commonly used as a marker of colonic SCFA production. The dietary intake was based on a thoroughly prepared food frequency questionnaire and judged as valid, although registration of the nutrient intake is afflicted with uncertainty. A more detailed and complete analysis of the faecal microbiome composition could have given other results. The metabolic and inflammatory changes and changes in other variables were incompletely recorded, and these results are therefore less reliable. The use of antibiotics, which were not registered, might have influenced the microbiota and their metabolites. Because the clinically important results were highly significant, it is unlikely that correcting for multiple testing, which was not performed, would have changed the main conclusions.

## 5. Conclusions

This study in subjects with morbid obesity showed significant changes in faecal SCFA levels after a combined conservative and surgical weight-loss intervention. The total amount of SCFA was reduced, the total and relative amounts of the main straight SCFA (acetic-, propionic-, and butyric- acids) were reduced, and the total and relative amounts of the branched SCFA (isobutyric-, isovaleric-, and isocaproic- acid) were increased. These changes indicate an alteration in the balance of saccharolytic and proteolytic fermentation toward a proteolytic fermentation pattern with unfavorable health effects. There were significant associations between the amount of total and straight SCFA and the diet. No associations were seen with the metabolic markers and the faecal microbiome composition markers. Although the metabolic changes after bariatric surgery are complex and only partly characterized in this study that also had other limitations, the recommendation of a carbohydrate-rich diet after bariatric surgery in order to augment the saccharolytic- and reduce the proteolytic- fermentation seems to be reasonable clinical advice.

## Figures and Tables

**Table 1 nutrients-12-00802-t001:** The total and relative amounts of short-chain fatty acids (SCFA) at inclusion and changes after the weight-loss interventions. Analyzed with mixed model adjusted for point of time and the means of age and gender.

Dependent Variable	At Inclusion		Change T2 ^4^ Minus T1 ^3^		Statistics (*p*-Value)
	Mean	95% CI	Mean	95% CI	
Total SCFA ^1^	36.96	33.34; 40.59	−5.61	−10.43; −0.79	**0.023**
Acetic acid ^1^	20.28	18.37; 21.18	−3.78	−6.33; −1.23	**0.004**
Acetic acid (proportion ^2^)	55.14	53.76; 56.52	−1.66	−3.70; 0.38	0.109
Propionic acid ^1^	6.49	5.73; 7.26	−1.03	−2.05; −0.01	**0.048**
Propionic acid (proportion ^2^ )	17.40	16.49; 18.32	−0.42	−1.58; 0.72	0.461
Butyric acid ^1^	7.23	6.35; 8.12	−1.31	−2.50; −0.13	**0.031**
Butyric acid (proportion ^2^)	18.97	17.89; 20.04	−0.38	−1.77; 1.00	0.582
Valeric acid ^1^	1.01	0.86; 1.16	0.01	−0.20; 0.22	0.904
Valeric acid (proportion ^2^)	2.68	2.42; 2.94	0.56	0.21; 0.91	**0.002**
Caproic acid ^1^	0.31	0.23; 0.40	−0.06	−0.17; 0.06	0.353
Caproic acid (proportion ^2^)	0.79	0.56; 1.02	0.17	−0.14; 0.47	0.281
Isobutyric acid ^1^	0.70	0.60; 0.81	0.22	0.08; 0.36	**0.002**
Isobutyric acid (proportion ^2^)	2.01	1.78; 2.22	0.90	0.55; 1.24	**< 0.001**
Isovaleric acid ^1^	1.02	0.87; 1.18	0.36	0.15; 0.57	**0.001**
Isovaleric acid (proportion ^2^)	2.94	2.60; 3.28	1.41	0.96; 1.86	**< 0.001**
Isocaproic acid ^1^	0.00	0.00; 0.00	0.00	−0.00; 0.00	0.753
Isocaproic aicd (proportion ^2^)	0.00	−0.00; 0.01	0.0	−0.01; 0.01	0.803
Straight SCFA ^1,5^	33.93	30.60; 37.26	−6.11	−10.59; -1.63	**0.008**
Straight SCFA ^5^ (proportion ^2^)	91.60	90.79; 92.41	−2.77	−3.79; −1.75	**<0.001**
Branched SCFA ^1,6^	1.72	1.46; 1.97	0.59	0.25; 0.93	**0.001**
Branched SCFA ^6^ (proportion ^2^)	4.95	4.40; 5.50	2.31	1.54; 3.08	**<0.001**

^1^ mmol/kg wet weight. ^2^ The proportion is given as the percentage of total SCFA. **^3^** T1: At inclusion. **^4^** T2: 6 months after surgery. ^5^ The sum of acetic-, propionic-, and butyric- acids. ^6^ The sum of isobutyric-, isovaleric-, and isocaproic- acids.

**Table 2 nutrients-12-00802-t002:** The amounts of nutrients (absolute and relative), blood biomarkers and the faecal microbiota at inclusion and changes after the weight-loss interventions. Mixed model adjusted for the means of age and gender.

Dependent Variable	Inclusion		Change T2 ^4^ Minus T1 ^3^		Statistics (*p*-Value)
	Mean	95% CI	Mean	95% CI	
Nutritional variables					
Energy total (KJ)	10662	9647; 11678	−4404	−5359; −3451	**<0.001**
Total food intake (g)	4971	4496; 5447	−1410	−1952; −869	**<0.001**
Protein (g)	112	100; 124	−37	−44; −31	**<0.001**
Protein (energy-%)	18.2	17.5; 19.0	2.0	1.0; 3.0	**<0.001**
Fat (g)	100	89; 111	−44	−54; −34	**<0.001**
Fat (energy-%)	34.2	32.8; 35.6	−0.7	−2.6; 1.1	0.435
Carbohydrates (g)	275	247; 302	−116	−151; - 80	**<0.001**
Carbohydrates (energy-%)	44.1	42.5; 45.8	−1.8	−3.9; 0.4	0.102
Sugar (g)	46	32; 59	−26	−46; −6	**0.011**
Sugar (energy-%)	6.4	5.1; 7.7	−1.9	−3.7; −0.2	**0.032**
Starch (g)	134	124; 145	−65	−78; −53	**<0.001**
Starch (energy-%)	21.9	20.6; 23.1	−2.7	−4.3; −1.0	**0.002**
Fibre (g)	35	32; 37	−12	−15; −10	**<0.001**
Fibre (energy-%)	2.8	2.6; 3.0	0.2	−0.1; 0.4	0.139
NNS (units) ^1^	8.0	6.0; 10.0	−2.8	−5.2; −0.5	**0.020**
Blood biomarkers					
CRP	6.9	6.0; 7.8	−5.0	−6.1; −4.0	**<0.001**
HbA1C	6.0	5.7; 6.2	−0.7	−0.9; −0.5	**<0.001**
Zonulin (ng/ml)	65	59; 70	−35	−44; −27	**<0.001**
Microbiota					
Dysbiosis Index (score 1–5)	2.7	2.5; 3.0	1.4	0.9; 1.9	**<0.001**
Firmicutes (mean score) ^2^	−0.00	−0.04; 0.04	0.16	0.09; 0.22	**<0.001**
Bacteroidetes (mean score) ^2^	0.43	0.37; 0.50	−0.08	−0.19; 0.03	0.151

^1^ NNS: Non-nutritive sweeteners. One unit of NNS was 100 mL beverage with NNS or two tablets/teaspoons of NNS. ^2^ Score range: −3; 3. **^3^** T1: At inclusion. **^4^** T2: 6 months after surgery.

**Table 3 nutrients-12-00802-t003:** Associations between the SCFA levels and the nutrients, biological markers and the microbiota composition markers analyzed with mixed model adjusted for the point of time and the mean of age and gender.

Independent Variables	Dependent Variables					
	Total SCFA (mmol/kg Wet Weight)		Straight SCFA ^1^ (mmol/kg Wet Weight)		Branched SCFA ^2^ (mmol/kg Wet Weight)	
	B (95% CI)	*p*-Value	B (95% CI)	*p*-Value	B (95% CI)	*p*-Value
Nutritional variables						
Energy total (KJ) ^3^	1.10 (0.14; 2.05)	**0.026**	1.06 (0.18; 1.94)	**0.019**	0.00 (−0.06; 0.08)	0.803
Total food intake (g) ^3^	1.55 (−0.16; 3.12)	0.052	1.14 (−0.27; 2.85)	0.054	0.07 (−0.5; 0.18)	0.246
Protein (g)	0.16 (0.06; 0.26)	**0.002**	0.15 (0.06; 0.24)	**0.002**	0.00 (−0.00; 0.01)	0.201
Fat (g)	0.13 (0.04; 0.21)	**0.004**	0.12 (0.04;0.20)	**0.003**	0.00 (−0.00; 0.01)	0.635
Carbohydrates (g)	0.01 (−0.01; 0.04)	0.350	0.01 (−0.01; 0.04)	0.299	−0.00 (−0.00; 0.00)	0.779
Sugar (g)	−0.03 (−0.07; 0.02)	0.305	−0.02 (−0.07; 0.02)	0.325	−0.00 (−0.01; 0.00)	0.379
Starch (g)	0.08 (0.01; 0.15)	**0.027**	0.08 (0.01; 0.14)	**0.018**	0.00 (−0.00; 0.01)	0.960
Fibre (g)	0.23 (−0.08; 0.54)	0.146	0.23 (−0.06; 0.51)	0.120	−0.00 (−0.02; 0.02)	0.984
NNS (units) ^4^	−0.14 (−0.50; 0.23)	0.460	−0.11 (−0.45; 0.22)	0.501	−0.00 (−0.01; 0.01)	0.620
Blood biomarkers						
CRP (mg/L)	0.27 (−0.39; 0.92)	0.426	0.25 (−0.35; 0.86)	0.409	0.00 (−0.05; 0.05)	0.977
HbA1C (%)	−1.48 (−3.93; 0.97)	0.234	−1.45 (−3.70; 0.80)	0203	−0.01 (−0.18; 0.17)	0.932
Zonulin (ng/ml)	-0.02 (-0.12; 0.08)	0.672	−0.02 (−0.11; 0.07)	0.669	0.00 (−0.01; 0.01)	0.718
Microbiota						
Dysbiosis Index (score: 1 to 5)	0.27 (−2.14; 2.69)	0.822	0.19 (−2.04; 2.43)	0.864	0.10 (−0.07; 0.27)	0.237
Firmicutes (score: -3 to 3)	−12.4 (−29.8; 4.9)	0.159	−11.2 (−27.2; 4.8)	0.169	−0.80 (−2.00; 0.40)	0.190
Bacteroidetes (score: -3 to 3)	−3.24 (−13.20; 6.72)	0.521	−2.63 (−11.82; 6.56)	0.572	−0.47 (−1.14; 0.21)	0.173

^1^ The sum of acetic-, propionic-, and butyric- acids. ^2^ The sum of isobutyric-, isovaleric-, and isocaproic- acids. **^3^** The B-values with CI are given as x 10 ^-3^.**^4^** NNS: Non-nutritive sweeteners. One unit of NNS was 100 mL beverage with NNS or two tablets/teaspoons of NNS.

**Table 4 nutrients-12-00802-t004:** Associations between changes in the SCFA levels and changes in nutrients, blood biomarkers and faecal microbiota composition markers (linear regression adjusted for age and gender).

Independent Variables	Dependent Variables					
Changes	Changes in Total SCFA (mmol/kg Wet Weight)		Changes in Straight SCFA ^1^ (mmol/kg Wet Weight)		Changes in Branched SCFA ^2^ (mmol/kg Wet Weight)	
	B (95% CI)	*p*-Value	B (95% CI)	*p*-Value	B (95% CI)	*p*-Value
Nutritional variables						
Energy total (KJ)	0.000 (−0.001; 0.002)	0.605	0.001 (−0.001; 0.002)	0.535	0.000 (0.000; 0.000)	0.161
Total food intake (g)	0.001 (−0.002; 0.004)	0.497	0.001 (−0.002; 0.004)	0.495	0.000 (0.000; 0.000)	0.895
Protein (g)	0.168 (−0.083; 0.418)	0.184	0.166 (−0.065; 0.397)	0.155	−0.004 (−0.022; 0.015)	0.681
Fat (g)	0.079 (−0.093; 0.252)	0.359	0.080 (−0.079; 0.239)	0.315	−0.007 (−0.019; 0.006)	0.272
Carbohydrates (g)	−0.001 (−0.046; 0.045)	0.970	0.001 (−0.041; 0.043)	0.965	−0.002 (−0.005; 0.001)	0.205
Sugar (g)	−0.035 (−0.099; 0.029)	0.280	−0.032 (−0.092; 0.027)	0.282	−0.002 (−0.007; 0.003)	0.425
Starch (g)	0.077 (−0.067; 0.221)	0.287	0.086 (−0.046; 0.218)	0.196	−0.012 (−0.22; -0.002)	0.019
Fiber (g)	0.357 (−0.258; 0.972)	0.249	0.369 (−0.199; 0.936)	0.197	−0.002 (−0.067; 0.023)	0.324
NNS (units) **^3^**	−0.125 (−1.094; 0.844)	0.796	−0.119 (−1.014; 0.776)	0.790	−0.025 (−0.095; 0.046)	0.485
Blood biomarkers						
CRP (mg/L)	0.779 (−0.298; 1.856)	0.153	0.680 (−0.316; 1.675)	0.176	0.059 (−0.017; 0.136)	0.127
HbA1C (%)	0.776 (−4.444; 5.996)	0.766	0.575 (−4.239; 5.389)	0.811	0.165 (−0.205; 0.535)	0.373
Zonulin (ng/mL)	−0.035 (−0.191; 0.120)	0.651	−0.038 (−0.182; 0.105)	0.596	0.005 (−0.006; 0.016)	0.384

**^1^** Changes in the sum of acetic-, propionic-, and butyric- acids. **^2^** Changes in the sum of isobutyric-, isovaleric-, and isocaproic- acids. **^3^** NNS: Non-nutritive sweeteners. One unit of NNS was 100 mL beverage with NNS or two tablets/teaspoons of NNS.

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
