# Peer review of "Changes in Faecal Short-Chain Fatty Acids after Weight-Loss Interventions in Subjects with Morbid Obesity"

_nutrients, 2020, doi:10.3390/nu12030802_

Round 1
Reviewer 1 Report
Please see attached file for reviewer comments.

Author Response
Reviewer 1
Comments to Authors on: Changes in Faecal Short-Chain Fatty Acids after Weight-Loss Interventions in Subjects with Morbid Obesity
“The primary aim was to study faecal SCFA in subjects with morbid obesity and changes in SCFA after a combined conservative and surgical treatment, and secondarily, to study associations between SCFA and the diet, the faecal microbiome composition and some metabolic and inflammatory biomarkers (HbA1c, CRP and s-zonulin).”
This was an interesting manuscript to review with a topic of clinical importance. More needs to be known about the impact common bariatric surgeries for weight loss have on patients, particularly as more becomes known about the influence of the gut microbiome on health.
Introduction
Given the aim of this study, this reviewer believes the introduction could be strengthened by adding some information about the impact the type of surgeries included have on the gut microbiome. This would help with the justification for the research as well. For example, Wagner et al 2018 published an integrative review which suggests bariatric surgery, of the types included in this manuscript, may increase Bacteroidetes and Proteobacteria and decrease Firmicutes (opposite of what the authors found). This is a more recent reference than reference #3 in the manuscript.
Response: The paper by Wagner et al. has been added to the reference list and referred to three times. The discrepancy between the results in the study by Wagner and this study has been added to the discussion.
Materials and Methods
This reviewer believes there are several places throughout this section where some clarification is needed in order for readers to better understand the study. As written, some items appear to be in conflict.
Lines 46-50 2.1 Study design
Later in the manuscript the authors mention that the 6 months of conservative treatment followed by surgery is a standard procedure for bariatric surgery for weight loss. It may help to strengthen the manuscript by stating this here in this section. A six-month or more conservative treatment time frame may not be the same across countries: it is not a requirement in the U.S. according to my knowledge. For readers not familiar with bariatric surgery, it may also be beneficial to indicate why there was a conservative treatment period: e.g., to help patients adapt to lifestyle changes that will be needed after surgery; and/or determine his/her ability to comply and ultimately be successful with weight loss.
Response: The comments have been added to the section “Study design”
Line 62 2.3 Interventions
Crispbread diet – for nonEuropean readers it would be beneficial to detail what this diet is as this reviewer has no idea what a crispbread diet is other than knowing there are different kinds of crispbread crackers.
Response: A reference to the crispbread diet has been added (line 64).
This does lead to a bit of concern about the lack of information on diet during the six-month conservative period other than the amount of kJ of energy. What about the level of protein, carbohydrates, fat and fibre? More information on the conservative treatment period would help to strengthen the manuscript.
Response: Except for the period with crispbread diet, the lifestyle and dietary advice were rather general.
Lines 65 – 98 2.4 Variables
This is the section of the Materials and Methods that this reviewer feels need the most clarification to help with the readability of the manuscript and to more easily understand the research which was performed. There are also points of concern.
Line 66-67 States the following variables were collected at inclusion (T1) and six months after bariatric surgery (T2)
Why was data not collected after the six-month conservative treatment period? This seems like some key missing information to help tease out the influence of the bariatric surgery.
Response: At the visit immediately prior to the operation, unfortunately it turned out to be nearly impossible to collect faecal samples and fill in a long food frequency questionnaire because the subjects were so focused on the operation.
This also appears to be in conflict with what is stated in Lines 93-98 of this section. Why were faecal samples collected at 12-months for some participants, but not others? There is no justification given for this time point. Six months is a long time in between and if there is no dietary information, how accurate and reliable are the SCFA data? The authors did state that in subjects with two analyses, there were no significant differences between results. However, of major concern is there is no information on the sample size for 12-month collections or even the 6-month collections (implied by Line 96). “In subjects with only one measurement, the results of the available test 6 or 12 months after surgery were used. In subjects with two analyses, the mean values of the two tests were used.” If samples were collected from all subjects at 6 months, then why the need to use a mean value of 6 and 12 months for only some of the subjects? This part is not clear and seems in this reviewer’s opinion to provide some unsound statistical analysis. This may be an issue of language and wording, but it does need clarifying.
Response: The subjects had a less extensive follow-up visit after 12 months, and they were asked to bring along a faecal sample if they had forgotten the one at the 6-months visit. Some had faecal samples after both 6 and 12 months. 6 months was preferred because the subjects are in a stable phase after the surgery, as they are after 12 months. The number of subjects available after 6 months has been added (line 133).
Line 75 – the authors indicate that dietary habits were assessed with a self-reported food frequency questionnaire. Was this data collected at T1 and T2? Again, why not prior to surgery, as bariatric surgery patients need to have a significant change in diet after surgery. How much did diet change from T1 to just before surgery?
Response: Please see the response above.
- Results
Lines 113-123 3.1 Subjects
Some additional information on the included subjects would be beneficial and may help to better interpret the obtained results.
There are also some discrepancies that need clarifying and are of major concern.
The inclusion criteria included a designation of morbid obesity, which was a BMI > 40 or > 35 with obesity-related complications. What was the n for each group? With a mean BMI of almost 42, what percentage of patients had Type 2 Diabetes or other comorbid conditions that could potentially impact gut health? What pharmacotherapies were these subjects on and were there any that could impact gut motility and gut health? No sample sizes are given in any of the tables and there is various wording throughout the manuscript that indicates samples sizes were not equal across variables.
Response: In our opinion, the mean BMI with SD is more important than the number in each group. All comorbidities, pharmacotherapies etc were not registered in detail. This has been added as a limitation (lines 277-278). It is correct that some variables were missing at T1 and other at T2, but to add the number to each line in the tables is unnecessary. The statistics “mixed model” takes care of the missing values.
The authors indicated that data from 111 subjects was included in the analyses. But, later on it is stated that only 90 had bariatric surgery. The authors also state that weight-loss was significantly higher in subjects who underwent the Roux-en-Y procedure versus those who underwent the gastric sleeve. What about the 21 who had no surgery? What was their weight-loss? Without the invasive surgery, it seems to this reviewer to be scientifically and statistically unsound to include their data with subjects who had a highly invasive surgery. These subjects would have significant physiological and immunological changes that have the potential to impact gut health and the gut microbiome. The Roux-en-Y patients could have issues with dumping syndrome, again potentially impacting the SCFA results. No information is given on complications associated with surgery and during the 6 months after surgery. Could this have impacted the SCFA results?
Response: Thanks for this valuable comment that actually made us aware of an error in the manuscript; the patients not operated were included in some of the analyses. This part of the paper has been rewritten. We have clarified that only patients with an operation (no = 90) were included in the analyses, and the analyses have been corrected. The incomplete registration of complications has been mentioned as a limitation (line 278).
It is indicated that there was a decrease in BMI of 0.15 per year of increasing age; was this for both men and women? Were there no gender effects? How much weight was lost after the six-month conservative treatment period? The sample size was 17% men and 83% women, so was the reduction in BMI consistent across both men and women, especially as BMI was noted to be higher in men at the start of the study.
Response: The weight loss was adjusted for gender. The difference between the genders was not statistically significant. The information about the weight-loss after conservative treatment was omitted because the entire study focuses on the effects of the combined conservative and surgical treatment.
- Discussion
Lines 179-180 Why? If a timepoint had been added at the end of the six-month conservative treatment period and just before surgery, would that not have helped determine the impact of the surgery on all variables? Lines 183 – 185 do indicate this is a standard protocol for treatment. The authors do indicate in Lines 207 – 209 that is a potential limitation of the study.
Response: We have explained above the reasons for studying the combined treatment and mentioned the limitation.
Line 181 – should there be a citation for the in press manuscript?
Response: The wording has been changed to “submitted by one of the authors”
Lines 197- 200 mentions the potential impact of diabetes, but yet there is no mention of the percentage of subjects with diabetes in this study unless the authors are lumping that into organic GI disorders.
Response: The comment about diabetes has been deleted.
Lines 219-220 – This sentence is unclear. The changes in this study that are presented in this study as results were from a previous study?
Response: The wording has been changed (line 240).
Lines 224 – 227 The authors do note a limitation about phyla which is good.
It is not until near the end of the discussion that a serious point of concern arises that was not mentioned in the Materials and Methods. Lines 228 and following.
The authors state: “These factors were incompletely recorded in this study.” This is in reference to CRP, HbA1c and zonulin. If this is a wording issue, then it needs to be rectified. If it is an issue with sample size of the various variables, then is of significant concern for scientific and statistical soundness. This reviewer is left questioning the lack of given sample size for the variables. This is stated again in Line 263 and the authors even state “these results are therefore less reliable.”
Response: The wording has been improved (lines 253-255).
Lines 250 – 253 – What implication does this recommendation have for weight loss and preservation of lean body mass after surgery? Patients who adhere to the dietary plan and physical activity after surgery can lose significant amounts of weight within a short time frame. Thus, the need for increased protein to help preserve lean body mass.
Response: The recommended moderate dietary change will not result in protein shortage.
Lines 257 – 260 the authors do note the limitations of using faecal SCFA versus colonic.
- Conclusions
Lines 274 and following. This reviewer is not in complete agreement that the data support the stated conclusions, especially as related to clinical advice for bariatric surgery patients.
Response: No comment about the agreement.
So wt loss greater with RnY but no change in SCFA
Reviewer 2 Report
The manuscript titled ‘Changes in faecal short-chain fatty acids after weight-loss interventions in subjects with morbid obesity’ investigates effects of combined conservative intervention and surgical weight-loss treatments on gut microbiome, fecal short chain fatty acids profile and inflammation marks in obese patients. The scale of the study and numbers of patients enrolled are impressive. One interesting finding of the study is that following treatments, changes in SCFAs are associated with a shift in the balance of saccharolytic and proteolytic fermentation toward a proteolytic fermentation pattern. However, there were also concerns,
Major:
One major concern regarding all sections of the paper is that the description is not sufficient.
For Introduction: for instance, line 29-36, the major SCFAs, i.e. acetic acids, propionic acids and butyric acids, play major roles in physiology, and differently. This should be introduced first, and the known association with microbiota. Only then, the changes found in obesity and/or with interventions, increased or decreased and possible mechanisms would make sense. The first time the authors mentioned butyrate serves as fuel for colonocyte was in discussion, even then, it stated SCFA, while the reality is that for colonocyte growth, butyrate is a unique energy source.
For material and methods, line 72-74: not enough information is provided, what methods, what kits, cat. number etc.. For all these parameters that evaluated, at what time point samples were taken, from the experimental design, it could be couple of interesting time.
For results: the authors provide big tables and stated details shown in table is overly simplified.
For Table 1, line 132-137:
- I don't get why the result of energy intake, blood test and fecal microbiota should be in the same results section? what's the logic here?
- Please remove the footnotes and place them which places they belong to in the table. It makes the table hard to read and please modify the table a bit, for instance, add lines to break it into different parts, maybe according to your way of classifying them total vs. total straight vs. total branched. etc.
- The two measurements, straight SCFA 33.01 and branched SCFA 1.75 add up as 34.76, what else was there to mount total SCFA 36.02?
Minor
Line 24: as dietary intervention, which is part of the conservative intervention is not described in previous sentences, here, the changes of SCFA are associated with diet conclusion came a bit sudden. Specify the conservative intervention.
Line: 83-86, shouldn’t this be results?
Line 87-88: what does it mean? That the bacteria do not represent the entire phyla.
Line 141-142: you need to specify what all these markers and metabolic syndromes are and add reference. and spell it out!
Line: 144: I do not get this, first, type of surgery is not included in 3.3 subheading. Second, is this mean two types of procedure do not differ between each other, or they do not differ from the ones without surgery? I also wonder during the surgery, antibiotics must be used, right? That does not have any effect on micrbiota, or the effect was gone with time?
Line 181: In press from the authors or from other people? Provide more information.
Line 219: if the method does not allow the characterization of microbiota, could you justify why use it?
Line250: as straight SCFA as authors described accounted for 91% of total, whereas branched SCFA was around 5% of total, it is not called a shift of fermentation pattern from one another, at best, alterations.
Author Response
Reviewer 2
Major concerns:
Comment: One major concern regarding all sections of the paper is that the descriptions are not sufficient.
Response: The sections have been enlarged based on the reviewer’s comments, see below.
Comment: The introduction, lines 29-36.
Response: The reviewer prefers a detailed description of the favourable effects of the straight SCFA and unfavourable effects of the branched SCFA in the introduction. In our opinion, the introduction should be brief and to the point and give a concise overview of the field. Because the study shows an alteration in SCFA we find it important to discuss in detail the clinical consequences of the findings in the “discussion”. A detailed presentation of the effects of individual SCFA in the introduction indicates that the results are known. We prefer not to change the introduction but will, of course, revise the introduction if the Editor agrees with the reviewer.
Comment: Material and methods, lines 72-74.
Response: Information about methods have been added (new lines 76-78). No new information has been given about the time points for the sampling since this is written on line 68.
Comment: For results: ….. stated details in the table are overly simplified.
Response: According to guidelines for scientific papers, duplication of information should be avoided. The table gives the detailed results, and the text emphasizes the main and most interesting results. This way of presenting the results avoids duplicating and makes it possible to read only the text and understand the results and conclusions without an in-depth study of the tables. In addition, it gives the interested reader the possibility to find all the details in the tables.
Comment: Why are the results of the energy intake, the blood tests and the faecal microbiota in the same results section?
Response: The primary aim of the study was SCFA and changes in SCFA, and these results were therefore presented in one paragraph in the text and in one table. The energy intake, blood tests and analyses of the faecal microbiota were secondary aims and included in one paragraph and one table.
Comment: The footnotes and extra lines in the tables.
Response: The footnotes have been written and placed under the tables according to MDPI’s guidelines. Please let us know if we have misunderstood the guidelines. Extra lines have been added to the tables 2 – 4.
Comment: About the total amount of SCFA.
Response: Straight SCFA was defined as the sum of Acetic-, Propionic-, and Butyric acid, and branched SCFA as the sum of Isobutyric -, Isovaleric-, and Isocapronic- acid. The difference between the sum of straight and branched SCFA, and the total SCFA, is the sum of Valeric- and Caproic- acid.
Minor concerns:
Comment: Line 24: as dietary intervention, …….
Response: We have written: “The changes in SCFA were associated with the diet”. The changes in the diet are from inclusion to 6 months after surgery and include the changes during conservative treatment and the changes all subjects have to do after the surgical procedure. It is demanding to present it more precisely with only 200 words allowed in the abstract.
Comment: Line: 83-86. (in the previous manuscript)
Response: The new lines are 95-98. These lines are not the results but a description of the methods.
Comment: Line 87-88: (in the previous manuscript). What does it mean?
Response: New lines: 100-101. The method used for the analyses of the microbiota composition measured only parts of the entire phyla. The sentence has been modified.
Comment: Line 141-142 (in the previous manuscript)
Response: New lines 157-158. The markers have been specified in the revised manuscript.
Comment: Line 144
Response: New line 156. The type of surgery means either Roux-en-Y or gastric bypass. There was no comparison with subjects without surgery because subjects without surgery had no follow-up visits. The heading 3.3 has been changed as proposed by the reviewer. A comment about antibiotics has been added to the limitations (lines 289-290).
Comment: Line 181
Response: New line 200. It has been added: “In press by one of the authors”
Comment: Line 219.
Response: New lines 239-240. We have added: “The method ……. did not allow a precise characterisation…..” since the method only measured the relative abundance of 39 bacteria at different taxonomic levels.
Comment: Line 250.
Response: New lines 272 and 298. The word “shift” has been replaced by “alteration”.
Reviewer 3 Report
The authors investigated changes in plasma levels of short-chain fatty acids (SCFAs) six month after bariatric surgery (RYGB+SG) in 111 obese subjects. The intervention also include a caloric restriction diet before the operation so the effects reported do therefore captures both the effect of caloric restriction and the operation.
Overall, I find the topic of interest and I have no major challenge with the methodology used. Although, I find the choice of microbiota analyses somewhat limited (why not choose a more broad approach like next generation sequencing) and I am not entirely sure how and what these data contribute to.
I have some points that the authors might consider
1) The first sentence in the introduction: The gut microbiota and their metabolites, e.g. short-chain fatty acids (SCFA), have important health-related effects and have been associated with a wide range of disorders, I find a little biased as there are emerging studies showing that the microbiota has not relevance in metabolic diseases as an example. I would therefore suggest to downtone words like important.
2) The primary aim seems to include several subaims. One thing I find missing generally is some considerations on sample size hence power.
3) the majority included are female, why? I would strongly suggest this to be noted as an limitation
4) A table showing baseline characterics seems to be missing?
5) the data on nutrient (table 2) is for this reviewer likely to be a major limitation of the subsequent analyses. My point is that reporting of nutrient intake is typically associated with a large bias or what?
6) The interpration of data need to be done a little more carefully I think as there are so many metabolic parameters changing after weight loss and after bariatric surgery.
Author Response
Reviewer 3
Point 1: The introduction.
Response: The word “important” has been removed from line 30
Point 2: About aims and sample size/power
Response: Reviewer’s comment about the primary aim is correct. The sentence (lines 42-43) has been modified to show that the primary aim has two parts. No power calculation was performed during the planning of the study. The sample size was fixed by the available study population. To do a power calculation after having performed the study gives no meaning. A sentence about sample size and power has been added (lines 120-122).
Point 3: About females.
Response: There is a majority of females in most studies in subjects with morbid obesity and bariatric surgery, and all analyses have been adjusted for age and gender. Therefore, we judge the external validity of the study as high, and doubt that the female majority is a limitation. Nevertheless, the sentence has been rephrased (lines 279-280)
Point 4: Baseline characteristics.
Response: Baseline characteristics are given in the text lines 129 – 140 and in tables 1 and 2.
Point 5: Nutrient intake.
Response: The report of nutrient intake is always burdened with uncertainty. This has been added to the limitations (line 286).
Point 6: Interpretation of the data.
Response: The interpretation of our own data is always demanding; there is a tendency to “over-emphasize” the importance of the results. A sentence has been added to adjust the conclusion: Although the metabolic changes after bariatric surgery are complex and only partly characterised in this study that also had other limitation, …… (lines 303-305).
Round 2
Reviewer 1 Report
The author replies responded to my concerns and changes were made. Thank you.
Author Response
Dear Editor,
Reviewer 1 had no comments to the revised version of the manuscript.
Best regards
Per G. Farup
Reviewer 2 Report
The manuscript has been improved a lot. I still feel the introduction doesn't provide enough information for the reader and too general. For instance, to claim straight SCFA are good, and the isoform SCFA are bad. They all present under physiological conditions, play roles. It is the disruption, the alteration of SCFA pattern, the imbalance give rise to detrimental effects.
Author Response
Dear Editor,
The introduction has been changed (lines 34-40) according to the comments given by reviewer 2.
Best regards
Per G. Farup
Reviewer 3 Report
I have no further comments
Author Response
Dear Editor,
Reviewer 3 had no further comments.
Best regards
Per G. Farup